# How CSV and CSR Affect Organizational Performance: A Productive Behavior Perspective

**DOI:** 10.3390/ijerph17072556

**Published:** 2020-04-08

**Authors:** Kwang O. Park

**Affiliations:** Division of Business, Yeungnam University College, 170 Hyeonchung-ro, Nam-gu, Daegu 42415, Korea; kopark1021@ync.ac.kr; Tel.: +82-53-650-9333; Fax: +82-53-625-6246

**Keywords:** CSR, CSV, work engagement, productive behavior, OCB, innovative behavior, job performance

## Abstract

**Background:** This study aims to shed light on the mutually beneficial causal relationship between creating shared value (CSV) and corporate social responsibility (CSR) activities and how they affect productive behavior through work engagement. Many preceding studies showed that work engagement and organizational citizenship behaviors (OCB) play a major role in the relationship between CSV and CSR activities and the organization’s internal performance. This study classified product behavior into OCB, innovative behavior, and job performance, based on the literature review. **Methods:** The subjects of this study were companies listed in KOSPI, which is Korea’s representative securities market. The companies listed on KOSPI are Korea’s leading companies as designated by the Korean government and financial authorities based on industry representation, market representation, and liquidity. **Results:** This study supported many preceding studies that analyzed the causal relationship between CSV and CSR activities, as well as OCB. In addition, this study has significant implications for businesses since it presents the possibility of studying the relationships between various organizational performance factors such as innovative behavior and job performance. **Conclusions:** It is expected that this study will help companies find more effective ways to strengthen their competitive advantage from a theoretical and practical perspective.

## 1. Introduction

Recently, companies have experienced increased expectation and demand to play more active social roles in society. Due to this, many companies have shown a more avid interest in corporate social responsibilities (CSR) by making investments in CSR and related activities. 

Moreover, companies have recently begun searching for a practical means of achieving mutual growth for business and society. This is a change from the traditional CSR method of using company profits to support social programs [1]. Companies have realized that they must both promote the corporation’s sustainable development and use their corporate expertise to solve critical societal issues [2]. 

Given these circumstances, the concept of creating social value (CSV) has attracted more attention. Recognition that CSV must become a cycle of sustainable development that promotes the mutual growth of society and companies has increased [3]. Therefore, many companies have greatly strengthened CSV activities, including win-win growth and support for socially disadvantaged individuals or groups to participate in economic activities, among others. 

CSV and CSR have a positive effect on a company’s expansion and continued growth [4]. Job creation and the formation of a social community can also be supported. In other words, CSV and CSR are based on symbiotic relationships between companies and stakeholders in society. CSV and CSR concepts impose ethical, legal, economic, and discretionary responsibilities on corporate stakeholders [3]. 

As society’s demand for ethical values increases, not only price and quality, but also ethical values are increasingly influencing consumers’ criteria for consumption. According to the results of the Korea Chamber of Commerce and Industry’s ”Consumer Perception on Ethical Consumption” survey in 2012, 59.6% of the 509 respondents had participated in ethical consumption in the past year. In addition, 72.9% responded that they would be interested in purchasing products reflecting ethical values, and 55.2% said they would be willing to pay additional costs of 5% or less for ethical consumption. These results indicate that corporate social responsibility activities positively affect trust between the company and customers, and lead to a positive corporate image [5].

The evolution in CSR methods has encouraged interesting research topics. One such topic is the participation of employees, such as their organizational citizenship behaviors (OCB), in CSR activities. Specifically, employee participation in CSR activities improves loyalty toward the organization and affects turnover intention [6].

Based on the findings of previous research, CSR concepts can be summarized as follows. CSR can be seen as a corporate management strategy that exhibits the company’s ethics and morality and allows the CEO and other company members to participate in social issues with responsibility and sympathy toward society. Not just the CEO, but other employees can also advance CSR activities by forming social networks.

Many studies of CSV and CSR have focused mainly on corporate financial performance and overall organizational performance. However, there is a relative lack of research incorporating the relationship between the organizational behaviors of internal corporate members, that is, CSV, CSR, and the micro-organizational behavior within the companies [6]. 

A company’s CSV and CSR activities can both make organization members proud of the organization and enhance their loyalty to the organization [7]. Furthermore, CSV and CSR activities can increase the members’ commitment to the organization and to their individual work. Therefore, this study aimed to analyze how the members of an organization respond to CSV and CSR activities. 

By analyzing the relationship between CSV and CSR and an organization’s internal performance, this study focused on employees’ work engagement and productive behavior as components of the organization’s internal performance. As mentioned earlier, productive behavior was divided into OCB, innovative behavior, and job performance. These definitions are based on a literature review and created for the purpose of analyzing productive behavior in this study [8].

Many studies consider CSR, work engagement, and OCB separately. In this study, however, OCB, innovative behavior, and job performance were combined into one category: productive behavior [6]. This allows for the measurement of the organization’s internal performance in a more specific and persuasive manner. Generally, job performance is considered representative of productive output. However, recently, many studies have categorized OCB and innovative behavior as forms of productive behavior and analyzed their correlations with other variables. Productive behavior can be explained as organizational behavior by employees that positively contribute to achieving the goals of the organization. Numerous studies have used it to assess the performance of organizations [6,8].

Therefore, a major goal of this study was to examine whether corporate CSV and CSR activities improve work engagement and productive behavior. Accordingly, this study examined the causal relationships that exist between these factors and attempts to verify them empirically. Notwithstanding the importance of these relationships on organizational research, there has been little research on the relationships between CSV, CSR, work engagement, and productive behavior. This study, therefore, is significant in the field of organizational research. Based on this background, a conceptual model was derived, as shown in Figure 1.

This study attempted to answer the following two research questions: 1: Do CSV and CSR affect work engagement? 2: Does work engagement affect productive behavior? 

In order to answer these questions, research was conducted on companies listed on KOSPI, the Korean stock market. These are Korea’s leading companies as determined by the Korean government and financial authorities and are based on market representation, industry representation, and liquidity.

Corporate research has traditionally focused on US firms. This is because the United States plays an important role in the global economy and business world. While the role of the US economy in the world is clearly recognized, research from an international perspective is also needed. Some countries around the world have made tremendous progress with the advancement and development of IT and internet technologies. According to the UK Business Software Alliance (BSA), the IT Human Resource Index showed that Korea ranked second in the world, after the United States [9]. In a similar index released by the UK’s Economist Intelligence Unit (EIU), Korea ranked third, after the United States and Japan [9]. Thus, research from a global perspective is necessary in an increasingly global market. It is expected that results from this study will help companies find effective ways to strengthen their competitive advantage from a theoretical and practical point of view.

## 2. Materials and Methods 

### 2.1. CSV and CSR

In view of the increased expectations for companies’ CSR activities, CSR has become a corporate duty, rather than a choice. In 1953, Bowen (1953) [10] first presented the CSR concept from the manager’s perspective. He defined CSR as “the obligations of businessmen to pursue those policies, to make those decisions, or to follow those lines of action, which are desirable in terms of the objectives and values of our society”. 

Since then, CSR has attracted the attention of many researchers [11,12], resulting in varied discussions on its effectiveness. These conversations have included the direct relationship between CSR and financial performance, as well as how CSR affects variables such as corporate evaluation, product evaluation, purchase intention, positive word of mouth, and loyalty [13,14,15]. CSR is likely to interact with and affect other business activities, such as production and technology [14].

CSV has recently evolved into a concept designed to strengthen companies’ core competitiveness through the creation of social values; this is beneficial for both the companies and society [4]. Porter and Kramer (2006) [11] offer a new perspective that does not see the relationship between business and society as a zero-sum game. They claimed that the goal of CSR should be simultaneous achievement of social benefits and a company’s economic benefits, that is, “shared value.” In other words, CSR should be expanded by making the most of the company’s expertise and resources, allowing for the improvement of the economic and social conditions in a community by using the company’s resources and expertise [7]. 

“Shared value” is not about the redistribution of earnings already generated by a company; instead, it is a concept that addresses the contemporaneous expansion of social values and the company’s economic profits. An example of this is fair trade, in which companies paying higher prices for crops produced by small farmers result in increased profits for the farmers; this is a redistribution of the profits already created by companies rather than “shared value”. 

While the CSV approach may require initial investment and time, both businesses and society may benefit from greater economic value and strategic benefits if small farmers’ income can be increased using the company’s expertise and resources, and if companies can be supplied with higher quality raw materials [11].

### 2.2. Work Engagement

Work engagement has become one of the most actively discussed concepts in motivation theory. In contrast to job burnout, work engagement is a positive, energetic, immersed, and dedicated state of mind about work [16,17]. It consists of three elements: vigor, dedication, and absorption [18,19]. 

Vigor means willingness to put effort into one’s work and enduring difficulties. Dedication means taking pride in one’s work and challenging oneself by being actively involved in one’s work. Absorption means being deeply immersed in performing one’s job [18]. 

Recently, research is being actively conducted to identify the antecedent variables and outcome variables that will increase the work engagement of an organization’s members [19]. In addition, after Schaufeli et al. [17] developed a metric to measure work engagement, research on work engagement accelerated. 

Studies on work engagement can be divided into two groups [20]. First, there are studies that identify antecedent variables that can predict work engagement by treating it as an outcome variable. Second, studies have examined the results of work engagement by considering it as a predictor variable. Recently, interest in work engagement has increased in the field of organizational behavior; while there are many studies underway, empirical research is still lacking. Therefore, the antecedent variables and outcome variables of work engagement have not yet been revealed.

### 2.3. Productive Behavior

Productive behavior is defined as the behavior of an organization’s members that positively contributes to achieving the organization’s goals and objectives. It is a representative behavioral organization performance within an organization that can be classified into three dimensions: OCB, innovative behavior, and job performance [8].

While a productive behavior may be generated from the expectation of a formal reward, an organizational behavior is exhibited when members perform tasks that are not required in their official job description. Without loyalty and trust for the organization, it is unlikely that this behavior will be exhibited [21]. As work engagement can contribute attitudes that support CSV and CSR within the organization, productive behavior may also be measured by the behavioral performance within the CSR-friendly organization.

#### 2.3.1. OCB

The concept of OCB was first used by Smith et al. (1983) [22], in order to refer to the discretionary behavior of organization members that contribute to organizational effectiveness. In the 1960s, Katz (1964) [23] described three essential behaviors of organization members that allowed an organization to function effectively. First, members must be involved in the organization and remain in the organization. Second, members must be trusted to play their assigned roles. Third, members must engage in creative and voluntary activities that go beyond their set roles [24]. 

These concepts were well organized by Organ (1988) [25], one of the most noted researchers in the field of OCB research. He defined OCB as “individual behavior that is discretionary, not directly or explicitly recognized by the formal reward system, and that in the aggregate promotes the effective functioning of the organization”. 

Organ’s definition of OCB includes three key aspects. First, OCB implies arbitrary behavior that is not among the assigned duties and is the result of individual choice. Second, OCB exceeds what is required of the job. Finally, OCB contributes positively to the overall efficiency of the organization [26,27]. 

Podsakoff et al. (2009) [28] found through meta-analysis that OCB was related to many organizational performance factors (e.g., productivity, efficiency, cost savings, customer satisfaction, and revenue per unit) and reported that OCB was a causal variable for organizational performance. 

OCB is important in the current corporate environment because horizontal team activities based on creativity and teamwork are becoming commonplace. Employees may need to voluntarily complete tasks outside of their specified roles in order to improve their relationship with their colleagues, which can improve team performance [29].

#### 2.3.2. Innovative Behavior

In today’s rapidly changing business environment, innovation is a key resource and strategic means of differentiating itself from competitors and securing competitive advantage in a competitive society [30]. 

In addition, the efforts of an organization’s members are essential at a time when customer demands are becoming more diverse and competition is becoming fiercer. There may be many methods and factors involved in achieving innovation in manufacturing or service, but the most effective method is the promotion of “innovative behavior” in individual organization members. 

According to Farr and West (1990) [31], innovative behavior may be defined as the intentional introduction of new and useful ideas and processes by individual members to their assigned roles. Innovative behavior, which refers to the voluntary action of creating new ideas that will help improve the performance of the organization, is regarded as the best concept of innovation that encompasses innovation on an individual level [32,33].

Therefore, this study also views innovative behavior as a series of activities by members of the organization to apply new ideas to work in order to help improve their work performance and the performance of the organization. It is important to note that innovation on an individual level focuses on the leading role played by individual members in relation to tasks they are responsible for, which can be distinguished from the innovative behavior of the organization [33,34].

#### 2.3.3. Job Performance

The most common form of productive behavior in organizations is job performance. Job performance can be defined as actions that members take within an organization and that contribute to the organization’s purpose [21].

In other words, job performance is composed of actions that are officially evaluated by the organization that the members are responsible for, and that constitute the members’ duties within the organization. Job performance can be divided into in-role task performance and extra-role contextual performance [35]. 

In-role task performance refers to the functional aspects of the employees’ duties and encompasses the performance of technical and specific duties for the organization, such as those specified in the job description. 

Extra-role contextual performance refers to nontechnical abilities, such as passion for work, motivation, effective communication, teamwork, and leadership, among others. 

Murphy (2013) [36] divided job performance into four dimensions. The first is task-oriented behavior that accomplishes the main tasks associated with the job’s duties. The second is interpersonal behavior, which encompasses all interpersonal interactions that occur in the job. Third is downtime behavior, which can be described as nontask-related organizational behavior that affects job performance. The fourth is destructive/hazardous behavior, which can be divided into safety violations, absences, walkouts, and sabotage, among others [37].

### 2.4. Research Model and Hypothesis

The purpose of this study was to examine the mutually beneficial causal relationship between CSV and CSR activities on productive behavior through work engagement. This causal relationship was empirically analyzed through a structural equation model. Therefore, the concepts constructed in the research model of this paper were designed as shown in Figure 2; this was used to derive the research hypotheses. 

### 2.5. Hypothesis and Sample Collection

CSV strengthens CSR because CSV aims to create common values for business and society [7,14]. The CSV approach may require initial investment and time. However, the income of small farmers can be increased using a company’s expertise and resources, and companies can continue to receive high-quality raw materials. This means that greater economic value and strategic benefits may result for both businesses and society [11]. Therefore, based on the preceding studies, the following hypothesis was established. 

**Hypothesis** **1.**
*CSV will influence CSR.*


Many studies have confirmed that organizational members’ perceptions of the company’s CSV and CSR activities have a variety of effects on their attitudes [19]. In addition, there are preceding studies in which work engagement of organizational members occurred when they recognized the company’s CSV and CSR [16]. Therefore, this study expected that the recognition of CSV and CSR will have a positive relationship with work engagement; thus, the following hypotheses were established. 

**Hypothesis** **2.**
*CSV will influence work engagement.*


**Hypothesis** **3.**
*CSR will influence work engagement.*


In the same way that work engagement is a desirable organizational behavior, productive behavior is also regarded as an important employee behavior [8]. When members identify with their organization’s positive image, they will trust the organization and be loyal toward it. They will also perform their tasks well and complete voluntary tasks. This will result in productive behaviors that are necessary for the organization, such as performing one’s duties in a responsible manner [37]. In addition, members who recognize CSV and CSR activities of what they view to be sincere companies are more likely to conduct OCB, which is beneficial to the organization [28]. Preceding studies [32] showed that CSV and CSR activities of companies have a positive effect on organizational performance through innovative behavior. The following hypotheses were established based on the relationship between CSV, CSR, and productive behavior in the preceding studies. 

**Hypothesis** **4.**
*CSV will influence productive behavior.*


**Hypothesis** **5.**
*CSR will influence productive behavior.*


The most widely accepted definition of work engagement was developed by Schaufeli et al. (2002) [17], who defined it as “a positive, fulfilling, work-related state of mind that is characterized by vigor, dedication and absorption”. An employee’s work engagement not only anticipates the individual or group performance but also creates additional customer loyalty through employees’ work efforts [38]. In preceding studies [21,32], it was found that work engagement had a significant effect on productive behavior. Therefore, based on the preceding studies, the following hypothesis was established. 

**Hypothesis** **6.**
*Work engagement will influence productive behavior.*


To verify the aforementioned hypotheses and the research model, a survey based on the preceding studies was conducted as shown in Table 1. For a more systematic survey, the company personnel were informed of the purpose of this study, and questionnaires were sent for a pilot test.

The subjects of this study were companies listed in KOSPI, which is Korea’s representative securities market. The companies listed on the KOSPI are Korea’s leading companies, as designated by the Korean government and financial authorities based on industry representation, market representation, and liquidity. A total of 250 questionnaires were sent by phone, by mail, and in person, from June 2019 to August 2019, and 198 responses to the survey were received, showing a 79% response rate. A total of 195 responses were used for the analysis after excluding incomplete responses with missing values. Table 2 summarizes the profiles and demographics of the companies and respondents that participated in the study. Statistical analysis was performed using SPSS 24.0 (IBM, Armonk, NY, USA) and AMOS 24.0 (IBM, Armonk, NY, USA). 

Common method bias may exist as independent and dependent variables were measured from the same respondents. To verify the results, we conducted Harman’s single-factor test. The results revealed that one factor accounted for 43.27% of the total variance, which satisfies the general standard set at below 50%. We can therefore infer that there were no issues related to common method bias.

## 3. Results

Confirmatory factor analysis was performed to verify the validity. The number of factors was determined based on the criterion of eigenvalues greater than one, and a varimax rotation was used for analysis. While the size of the sample required for a particular significance varies, factor loadings required for significance are considered to be in the 0.50–0.55 range for a sample size of over 100 [39]. Therefore, in this study, factors were extracted with the criterion of factor loadings of 0.50. The results of the factor analysis showed that all factor loadings for CSV, CSR, work engagement, OCB, innovative behavior, and job performance exceeded 0.50, as presented in Table 3, indicating discriminant validity between measurement variables and convergent validity within the variables.

In addition, the suitability of the research model was verified through the second-order construct model in order to measure productive behavior that was formed through OCB, innovative behavior, and job performance. The structural equation model was used to verify the significance of the research hypotheses. To test for the validity of the research model, convergent validity and discriminant validity were checked. Convergent validity refers to the degree to which two measures of constructs that theoretically should be related are actually related, and uses the construct reliability (CR) and the average variance extracted (AVE). When CR is 0.7 or more and AVE is 0.5 or more, it is considered to have convergent validity. As shown in Table 4, all CR values were above 0.70, and all AVE values exceeded 0.50, ensuring convergent validity [39].

Discriminant validity tests whether constructs that are not supposed to be related are actually unrelated. The test for discriminant validity is (square root of AVE) > (correlation coefficient), that is, the square root of AVE between the variables must be greater than the correlation coefficient [39]. As shown in Table 5, AVE values were greater than the squared correlation coefficients, indicating that the discriminant validity of the correlation between variables in this study was secured. In addition, the absolute values of correlation coefficients of all concepts did not exceed the criterion of 0.85, indicating no multicollinearity problems between the constructs; thus, the discriminant validity of constructs was viewed to be generally secured.

In addition, a test for multicollinearity problems was conducted through the variance inflation factor (VIF) and tolerance (TOL) methods, as shown in Table 6. The results showed that multicollinearity was not a problem. Common criteria for judging multicollinearity are a TOL value of 0.3 or more and a VIF value of 10 or less.

Structural equation analysis was performed using AMOS 24. The fit statistics of this study were good, except for TLI, as shown in Table 7. As the indicators demonstrate, it was determined that the analysis could be continued under the current conditions [40].

### The Model Structure

Structural equation modeling can be divided into first- and higher-order constructs. In the higher-order construct models, some constructs are formed by subconstructs. In this study, OCB, innovative behavior, and job performance were significantly loaded into productive behavior, that is, the dependent variable, as shown in Figure 3. 

Hypothesis 1, that CSV will have a significant effect on CSR, was found to be significant. In the first-order construct model for detailed model verification, CSV also had a significant effect on CSR. This result was similar to the results of studies [7] that had emphasized the relationship between CSV and CSR. Therefore, it is judged that CSV, which is aimed at creating common values of business and society, will have a positive effect on everyone by strengthening CSR. 

Hypothesis 2, that CSV will have a significant effect on work engagement, was found to be significant. In the first-order construct model for detailed model verification, CSV had a significant effect on work engagement. In addition, Hypothesis 3, that CSR will have a significant effect on work engagement, was found to be significant. In the first-order construct model verification for detailed model verification, CSR had a significant effect on work engagement. These findings are similar to studies [19] that emphasized the relationship between CSV and CSR activities and the attitudes of members. Therefore, the company’s CSV and CSR activities are expected to have a positive effect on employee work engagement. 

Hypothesis 4, that CSV will have a significant effect on productive behavior, was found to be significant. In the first-order construct model for detailed model verification, CSV had a significant effect on productive behavior. In addition, Hypothesis 5, that CSR will have a significant effect on productive behavior, was found to be significant. In the first-order construct model for detailed model verification, CSR had a significant effect on productive behavior. This result is similar to that of studies [32] that emphasized the relationship between CSV, CSR, and productive behavior. Therefore, it is judged that the company’s CSV and CSR activities have a positive effect on organizational performance through innovative capabilities. 

Hypothesis 6, that work engagement will have a significant effect on productive behavior, was found to be significant. In the first-order construct model for detailed model verification, work engagement had a significant effect on productive behavior. This result is similar to that of studies [21] that emphasized the relationship between work engagement and productive behavior. Therefore, it is judged that work engagement has a positive effect on organizational performance through improved loyalty. 

Detailed results as well as the direct and indirect effects are presented in Table 8.

## 4. Discussion

The purpose of this study was to examine the mutually beneficial causal relationship between CSV and CSR activities on productive behavior through work engagement. Most of the preceding studies had focused on work engagement and OCB through studying the relationship between CSV and CSR activities and an organization’s internal performance. However, in this study, OCB, innovative behavior, and job performance were all grouped together into productive behavior. This study supported many previous studies that analyzed the causal relationship between CSV and CSR activities and OCB. In addition, this study has significant implications since it presents the possibility of studying the relationship between organizational performance and various factors such as innovative behavior and job performance, among others. In addition, comparative studies with existing studies were conducted. 

First, it was shown that CSV had a significant effect on CSR. In addition, both the first and second tests showed that CSV had a significant effect on CSR. These findings were similar to those of Yoon et al. (2006) [7]. This is because CSV aims to create a common value between business and society; thus, CSR needs to be strengthened. Of course, the CSV approach may require significant investment and time in the beginning. However, this will eventually bring greater economic value and strategic benefits to both business and society. 

Second, CSV and CSR had a significant effect on work engagement. In addition, both the first and second tests had a significant effect on work engagement. The results showed that CSV had an indirect effect on work engagement, as shown in Table 8. These findings were similar to those of Schaufeli et al. (2006) [19]. They suggested that organizational members’ perceptions of the company’s CSV and CSR activities have a significant effect on their attitudes. Therefore, it is judged that the awareness of CSV and CSR by members of the organization will have a positive relationship with work engagement. 

Third, it was shown that CSV and CSR had a significant effect on productive behavior. In addition, both the first and second tests had a significant effect on productive behavior. The results showed that CSV had an indirect effect on productive behavior, as shown in Table 8. These findings were similar to those of Agarewal et al. (2007) [32]. They suggested that members who are aware of the company’s CSV and CSR activities are more likely to do what is good for the organization. Therefore, it is judged that the company’s CSV and CSR activities will have a positive effect on organizational performance through innovation capabilities. 

Fourth, it was shown that work engagement had a significant effect on productive behavior. In addition, both the first and second tests had a significant effect on productive behavior. Therefore, the higher the work engagement, the higher the satisfaction, which will eventually lead to productive behavior. Work engagement has proven to be the most important factor in improving organizational performance. These findings were similar to those of Kim and Park (2017) [21]. Based on these findings, employees’ work engagements can increase corporate performance through improved loyalty. 

Even though this study made a number of positive contributions from an academic and practical perspective, there are some limitations in the contents and methodology. 

First, one of the most important variables in this type of organizational research is CEO characteristics. However, we were not able to consider organizational strategy based on this variable. Future research needs to conduct additional in-depth investigations on the effects of CEO characteristics and leadership style. 

Second, organizational performance was limited to the internal performance of the organization consisting of engagement and productive behavior in this study. However, it may be necessary to consider the external performance, such as the company’s financial performance (e.g., sales, profit growth, and cost reduction), which may be the company’s most important goal, corporate image enhancement, and brand value creation, among others. This suggests that a study of other dependent variables may be needed in the future. 

Third, full control in obtaining a representative sample that properly and accurately reflects all the pertinent characteristics of the population, e.g., in terms of company size and industry, was somewhat lacking. In the future, if sample representativeness can be improved through a breakdown of company size, industry, etc., it should be possible to obtain more meaningful results. 

Fourth, a cross-sectional study was conducted for empirical analysis in this study. However, in order to more accurately identify the causal relationships between the variables used in this study, longitudinal studies would be desirable. This would allow the causal relationships between the variables used in this study to be judged more accurately.

## 5. Conclusions

The implications from the academic and practical perspectives of this study are as follows. First, in researching the relationship between CSV and CSR and organizational performance, most of the previous studies focused on work engagement and OCB. In this study, however, the organization’s internal performance was set to work engagement and productive behavior, while productive behavior was categorized into three organizational performance factors; that is, OCB, innovative behavior, and job performance, based on a literature review, and the relationship with CSV and CSR was then analyzed. This study supported many preceding studies that analyzed the causal relationship between CSV and CSR activities and OCB. In addition, it has significant implications since it presents the possibility of studying the relationships between various organizational performance factors, such as innovative behavior and job performance. 

Second, companies can increase their employees’ attachment and loyalty to the organization through the management of their human resources, through CSV and CSR activities, and by strengthening work engagement to build a positive attitude in the employees. It will also be important to encourage the voluntary and desirable productive behavior of members of the organization. This study implies that it may be helpful to have employees participate in CSV and CSR activities in order to encourage productive behavior. 

Third, the causal relationship between CSV and CSR was analyzed, which was lacking in preceding studies. Although the organization’s CSV and CSR activities are antecedent factors that have many effects within the organization, no research thus far had systematically revealed the relationship between them, thus providing a new framework for research. 

## Figures and Tables

**Figure 1 ijerph-17-02556-f001:**
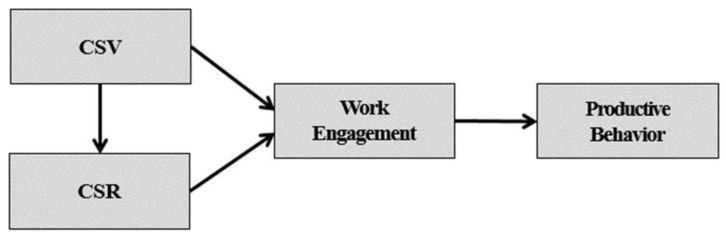
Conceptual research mode.

**Figure 2 ijerph-17-02556-f002:**
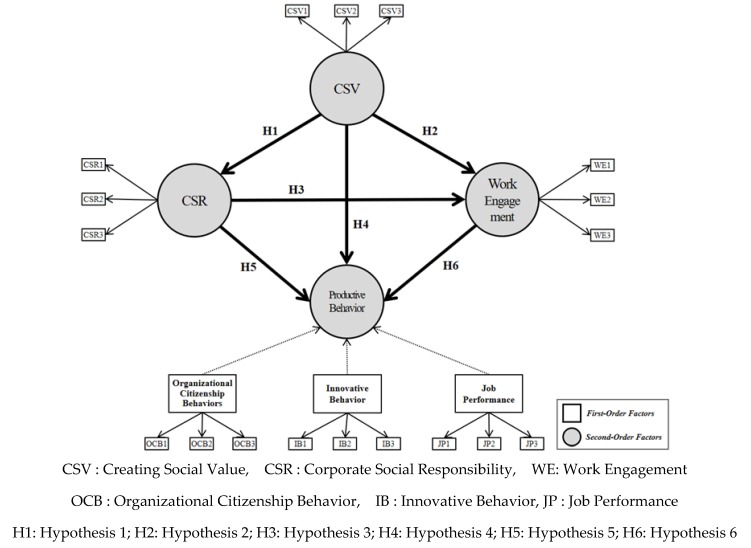
Research model.

**Figure 3 ijerph-17-02556-f003:**
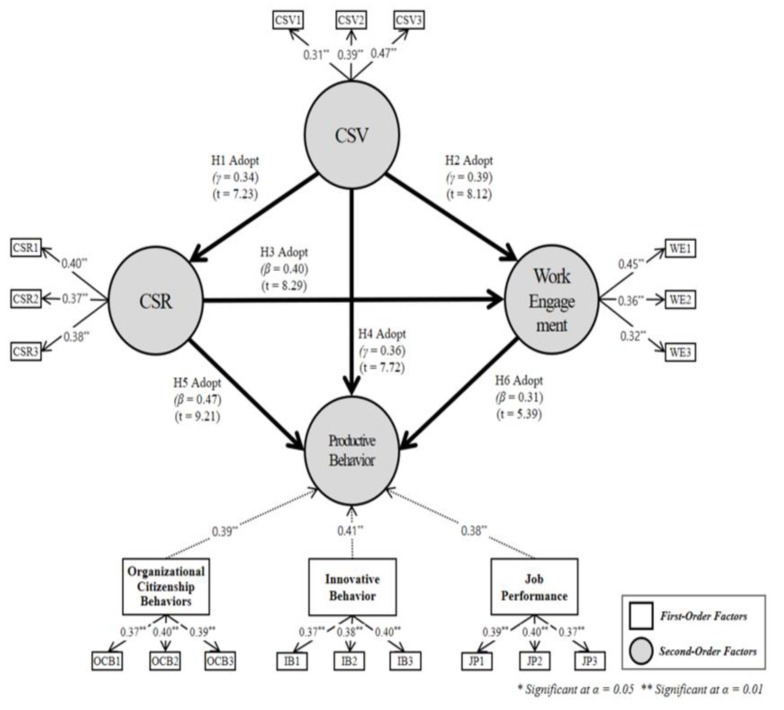
Results of hypothesis testing.

**Table 1 ijerph-17-02556-t001:** Research constructs and operationalization.

Construct	Items	References
Creating Social Value	“Win-Win” awareness by business and society	Porter and Kramer (2011) [26] Kim et al. (2010) [27]
	Using resources and capabilities for “Win-Win”	
	Contributing to social development and contribution activities	
Corporate Social Responsibility	Accepting the needs of society	Porter and Kramer(2011) [26] Yoon et al. (2006) [7] Sen and Bhattacharya (2001) [1]
	Exerting efforts to solve social problems	
	Improving social welfare or quality of life	
Work Engagement	Attributing value to work	Schaufeli et al. (2006) [19] Bhattacharya and Sen (2003) [16] Schaufeli et al. (2002) [17]
	Passionate about work	
	Work immersion	
OrganizationalCitizenship Behavior	Helping colleagues with problems encountered while working	Park (2019) [24] Podsakoff et al. (2009) [28]
	Making efforts to achieve results beyond standards	
	Loyalty and pride about the company	
Innovative Behavior	Proposing creative working methods	Agarwal et al. (2012) [32] Kheng and Mahmood (2013) [30]
	Generating ideas for solving work-related problems	
	Focusing on creativity, innovation, and challenges while working	
Job Performance	Performing given tasks perfectly	Kim and Park (2017) [21] Barrick and Mount (2005) [37]
	Completing tasks with a sense of responsibility	
	Meeting company expectations	

**Table 2 ijerph-17-02556-t002:** Profiles of companies and respondents.

	Frequency	Percent (%)
*Age of Respondent*		
30–40	95	49
40–50	68	35
Over 50	32	16
*Gender of Respondent*		
Male	118	61
Female	77	39
*Job Tenure of Respondent*		
1–5	75	38
5–10	63	32
Over 10	57	30
*Title of Respondent*		
Assistant manager	67	34
Manager	62	32
General manager	46	24
Executive director	20	10

**Table 3 ijerph-17-02556-t003:** Results of confirmatory factor analysis (each item is measured with a 5-point Likert scale).

Item	Job Performance	Work Engagement	CSR	Innovative Behavior	OCB	CSV
CSV1	−0.091	0.308	−0.002	0.019	0.132	0.751
CSV2	0.075	0.028	0.251	0.249	0.058	0.823
CSV3	0.131	0.26	0.066	0.163	0.266	0.788
CSR1	0.283	0.195	0.814	0.182	0.161	0.011
CSR2	0.145	0.279	0.845	0.127	0.138	0.133
CSR3	0.199	0.232	0.737	0.279	0.209	0.221
WE1	0.105	0.822	0.201	0.226	0.177	0.201
WE2	−0.002	0.836	0.28	0.215	0.056	0.14
WE3	0.148	0.825	0.175	0.047	0.091	0.24
OCB1	0.28	−0.057	0.048	0.191	0.777	0.173
OCB2	0.195	0.187	0.211	0.196	0.773	0.147
OCB3	0.091	0.204	0.209	0.144	0.872	0.129
IB1	0.132	0.236	0.369	0.779	0.202	0.141
IB2	0.195	0.145	0.238	0.874	0.183	0.138
IB3	0.327	0.176	0.033	0.781	0.238	0.228
JP1	0.855	0.069	0.108	0.263	0.191	0.049
JP2	0.872	0.016	0.195	0.265	0.094	0.007
JP3	0.807	0.173	0.271	0.003	0.277	0.039
CSV: Creating Social Value, CSR: Corporate Social Responsibility, Work Engagement OCB: Organizational Citizenship Behavior, Innovative Behavior, Job Performance						

**Table 4 ijerph-17-02556-t004:** Results of convergent validity.

Constructs	AVE	CR	Cronbach α
CSV	0.72	0.832	0.804
CSR	0.829	0.927	0.893
Work Engagement	0.786	0.846	0.829
OCB	0.792	0.872	0.861
Innovative Behavior	0.841	0.931	0.923
Job Performance	0.834	0.929	0.897

**Table 5 ijerph-17-02556-t005:** Results of discriminant validity.

Construct	CSV	CSR	Work Engagement	OCB	Innovative Behavior	Job Performance
**CSV**	0.849					
**CSR**	0.378 **	0.910				
**Work Engagement**	0.494 **	0.457 **	0.887			
**OCB**	0.417 **	0.369 **	0.361 **	0.890		
**Innovative Behavior**	0.344 **	0.412 **	0.469 **	0.418 **	0.917	
**Job Performance**	0.298 **	0.398 **	0.283 **	0.376 **	0.403 **	0.913

The shaded numbers in the diagonal row are square roots of the AVE. ** Significant at α = 0.01.

**Table 6 ijerph-17-02556-t006:** Variance inflation factor (VIF) and tolerance.

	Tolerance	VIF		Tolerance	VIF
CSV	0.740	1.352	CSR	0.676	1.479
Work Engagement	0.596	1.679	Dependent Variable: Productive Behavior		

**Table 7 ijerph-17-02556-t007:** Fit statistics for validating the measurement model.

Recommended Value		Measurement Model
**Fit statistic**	*X*^2^/DF (≤3.000)	2.720
	RMSR (≤0.050)	0.039
	RMSEA (≤0.080)	0.052
	AGFI (≥0.800)	0.824
	CFI (≥0.900)	0.918
	TLI (≥0.900)	0.891
	PGFI (≥0.600)	0.627

**Table 8 ijerph-17-02556-t008:** Coefficients of direct, indirect, and total effects.

		CSR	Work Engagement	Productive Behavior
CSV	Direct Effect	0.34	0.39 **	0.36 **
	Indirect Effect	-	0.12 **	0.13 **
	Total Effect	0.34	0.51 **	0.49 **
CSR	Direct Effect		0.40 **	0.47 **
	Indirect Effect		-	0.05
	Total Effect		0.40 **	0.52 **
Work Engagement	Direct Effect			0.31 *
	Indirect Effect			-
	Total Effect			0.31 *

* Significant at α = 0.05, ** Significant at α = 0.01.

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
