# Peer review of "How CSV and CSR Affect Organizational Performance: A Productive Behavior Perspective"

_ijerph, 2020, doi:10.3390/ijerph17072556_

Round 1

Reviewer 1 Report

Thank you for the opportunity to read your manuscript which I find interesting. Here are my suggestions to improve it. 

  1. You have to improve your introduction and include relevant cites that look at the relevant theories you are using such as CSR, employees CSR, and OCB. Some of these papers include: What drives employee involvement and turnover intentions: empirical investigation. Also, look at the effect improved firm reputation through its CSR towards both employees and external stakeholders and include relevant cites.
  2. Productivity of employees is the outcome variable but there is little theory that supports the argument, try to provide cites from previous research. 
  3. CEO characteristics matter for both internal and external CSR, please consider including these concepts in your literature along with relevant cites.

Author Response

Dear Reviewer:

I wish to resubmit an article for publication in International Journal of Environmental Research and Public Health, titled “How CSV and CSR Affect Organizational Performance: A Productive Behavior Perspective”.
The manuscript has been rechecked and appropriate changes have been made in accordance with the reviewers’ suggestions. The responses to their comments have been prepared and given below.

I would like to express my deep gratitude for your thoughtful suggestions and insights, which have enriched the manuscript and produced a better and more balanced account of the research. We hope that the revised manuscript is now suitable for publication in your journal. Please find below an overall list of all revisions made to the manuscript:

- Strengthen the introduction (Add theory and previous research, present the effects from the activities)
- Revise duplicated sentences
- Explain abbreviations to improve clarity
- Update with the latest research
- Strengthen the conclusion and future research tasks section
- Overall editing recommended by International Journal of Environmental Research and Public Health
MDPI (English editing ID: English-17637)

Thank you for your in-depth review. I look forward to hearing from you.

Respectfully ~ Best regards,

Reviewer 2 Report

The article has a good argument and is an interesting topic. I have some recommendations: 1. In the introduction, it is recommended to add the problems and challenges faced by such companies to generate value and develop CSR best practices for greater innovative production behavior and better economic results and financial results. Mention or theories that support this research, from the point of view of the sciences of psychology (productive behavior) and business sciences (CSR and performance). The question is what are the theories behind this study? 2. In the method, it is important that they more accurately justify the development of hypotheses, from a theoretical and empirical point of view. It is also important that you incorporate the questionnaire used in the appendix section. How did you control for non-response bias? They can use CMV through the Harman test (Variance of the common method-CMV). In the structural model, it is important that they explain the procedure used for the statistical analysis of first-order constructions and then the statistical procedure for second-order constructions (what are the guidelines?). It is recommended to incorporate at least 2 control variables (age, sex, position, etc.) in the proposed model to show an alternative model. 3. In the conclusions, it is important to make a better explanation of the theoretical and empirical implications. What is the main contribution of the study? 4. Carefully explain the limitations and future lines of inquiry.

Author Response

(The authors gave the same response as above.)

Round 2

Reviewer 2 Report

Your manuscript has been revised again and I see that it has been improved.

From my point of view the manuscript can be published subject to the comments and decision of the editor.